# Ego-Resiliency, Life Satisfaction and Symptoms of Anxiety and Depression Among Students of Pro-Health Faculties During COVID-19 Pandemic

**DOI:** 10.3390/healthcare13091008

**Published:** 2025-04-27

**Authors:** Karina Badura-Brzoza, Patryk Główczyński, Paweł Dębski, Małgorzata Dębska-Janus

**Affiliations:** 1Clinical Department of Psychiatry, Faculty of Medical Sciences in Zabrze, Medical University of Silesia, 40-055 Katowice, Poland; kbbrzoza@sum.edu.pl; 2Institute of Psychology, Humanitas University, 41-200 Sosnowiec, Poland; pdebski@sum.edu.pl; 3Institute of Sport Sciences, The Jerzy Kukuczka Academy of Physical Education, 40-055 Katowice, Poland; m.debska@awf.katowice.pl

**Keywords:** COVID-19 pandemic, students, ego-resiliency, life satisfaction, symptoms of anxiety and depression

## Abstract

**Background**: Ego-resiliency could play a protective role, especially in stressful situations. Such a situation is certainly the period of the COVID-19 pandemic. The beginning of the pandemic period was a source of significant stress for many people. Students, especially of medical faculties, were one of the social groups that could be particularly affected by the reorganization of everyday functioning. Effective coping with stress during this period could have been important for minimizing its negative effects. **Aim**: The aim of this study was to assess ego-resiliency as a factor enhancing life satisfaction and a protective factor against symptoms of anxiety and depression in a group of health students during the COVID-19 pandemic. **Methods and Material**: This study was conducted in the period from October 2020 to June 2021. The study group included 362 students of the medical faculty and 249 students of the Academy of Physical Education (APhE). The Ego Resiliency Scale (ER89-R12), the Life Satisfaction Scale (SWLS) and the Anxiety and Depression Scale (HADS) were used in this study. Among the statistical methods, the Mann–Whitney U test and Spearman’s rank correlation coefficient were used. **Results**: In the study group, medical students obtained an average score of 34.96 ± 5.19 points and APhE students obtained 36.49 ± 5.22 points on the ER-89-R12; the difference was statistically significant (*p* = 0.003). On the SWLS, the mean score was 23.65 ± 5.9 points for medical students and 22.35 ± 5.67 points for students of the APhE; the difference was also statistically significant (*p* = 0.005). In the assessment of anxiety, medical students obtained an average of 8.43 ± 4.34 points and students of the APhE 7.60 ± 4.27 points; the difference was statistically significant (*p* = 0.012). In the assessment of depression, medical students achieved 5.10 ± 3.77 points and students of APhE obtained about 4.77 ± 3.26 points; the difference was not statistically significant. There were significant, negative correlations in the scope of the results obtained on the ER-89-R12 with the results of anxiety and depression, and positive correlations with the results obtained on the SWLS scale for both groups. **Conclusions**: Life during the pandemic was assessed by students of pro-health faculties as moderately satisfactory. The severity of symptoms of anxiety and depression correlated negatively with life satisfaction. Ego-resiliency may be a factor enhancing life satisfaction and may be a protective factor against anxiety and depression symptoms.

## 1. Introduction

The term “resilience” comes from the Latin salire, which means spring, spring up; and resilire, which means spring back. There are two terms that can be found in the literature: resilience and resiliency. The first one is identified with the process of successfully overcoming negative phenomena and life events. The second means a personality property or a relatively permanent resource of an individual and is defined in Polish as mental resilience [1]. This understanding of resilience refers to Block’s theory of ego-resiliency [2], who treats it as a relatively permanent disposition, determining the process of flexible adaptation to constantly changing life requirements. Block refers to ego-control as the tendency to contain versus express impulse and ego-resiliency as the ability to modulate, in either direction, their characteristic level of ego-control, depending on situational demands. Thus, Block perceives it as a personality trait, important in the process of struggling with both traumatic events and events occurring in everyday life. Resilience stands in opposition to both a lack of control (impulsiveness) and an excess of it (rigidity). This means that resilience is related to the ability of an individual to adjust the level of control to their own capabilities and the requirements of the situation. A person with a high level of resilience exhibits an optimal level of mobilization, adequate to the actions taken [3]. In this context, mental resilience can be treated as a mechanism of self-regulation, including cognitive, emotional and behavioral elements [4]. The former are characteristic of beliefs and expectations and concern the perception of reality as a challenge, as well as one’s own competences. The emotional components of resilience are related to positive feelings and emotional stability. In turn, the behavioral components of resilience are expressed in the search for new experiences and the adoption of various and effective strategies for dealing with problems. Mental resilience will therefore foster perseverance and flexible adaptation to life requirements, facilitating the mobilization to take preventive actions in difficult situations, and will also increase the tolerance to negative emotions and failures [5]. The characteristics of people with high mental resilience are as follows: close, cordial interpersonal contacts, self-confidence, a calm manner, the ability to control emotions, perseverance in pursuing goals and achieving them, independence, autonomy and emotional stability. A person with a high level of resilience has a positive attitude towards life, is able to effectively deal with the challenges of life and surprising situations that require a flexible response [6]. They perceive the difficulties encountered more often as an opportunity to gain new experiences [7]. Mental resilience (ego-resiliency) could play a protective role, especially in stressful situations. Such a situation is certainly the period of the COVID-19 pandemic. The changes that have taken place in the surrounding reality, especially at the beginning of the pandemic period, have been a source of significant stress for many people. Students, especially of medical faculties, were one of the social groups that could be particularly affected by the reorganization of everyday functioning [8]. The sense of threat and uncertainty of tomorrow could significantly influence the change of lifestyle and generate anxiety and fear [9]. In addition, there were difficulties related to changes in academic learning, as well as direct contact with patients during clinical classes, where the potential risk of COVID-19 infection was much higher than in other circumstances. By pro-health faculties, the authors meant studies related to the science of the principles of observing healthy habits, learning about them and creating proper health attitudes, e.g., medicine or physical education studies. Polish medical students participated as volunteers performing swabs for the presence of the SARS-CoV-2 infection, and also took part in the treatment of patients in the so-called temporary hospitals. Functioning in the conditions of the need to exercise extreme caution, wear protective clothing and constantly changing guidelines regarding procedures were factors that generated tension and had an impact on health and satisfaction with life [9,10]. Effective coping with stress during this period could have been important for minimizing its negative effects, including long-term ones [11]. Ego-resiliency in this context may be a protective factor against the symptoms of tension related to changing living conditions, and at the same time may have a positive impact on its quality [12,13].

The aim of this study was to assess ego-resiliency as a potential protective factor against anxiety and depressive symptoms and having a positive relationship with life satisfaction in a group of health students during the SARS-CoV-2 virus pandemic.

## 2. Materials and Methods

In total, 930 students were invited to participate in this study. The study was conducted using university e-learning platforms. The study group included 362 medical students of Medical University of Silesia in Katowice aged 22.01 ± 2.90 years and 249 students of the Academy of Physical Education (APhE) in Katowice aged 21.40 ± 2.03 years. All respondents agreed to participate in the project. Before participation, students signed the special agreement form. Of invited students, 65.7% took part in the study. The sociodemographic characteristics of the respondents are presented in Table 1. Various differences between medical students and students at APhE are included in Appendix A.

The following tools and psychometric questionnaires were used to assess the parameters studied:Original demographic data questionnaire. This questionnaire was designed to obtain sociodemographic variables. It consisted of questions about gender, age, place of residence or level of education.The Ego Resiliency Scale (ER89-R12) is a tool used to measure mental resilience understood as a personality trait. Ego-resiliency is a dimension proving the human ability to dynamically, adequately and adaptively self-regulate, and therefore has a pro-health significance. This tool is a Polish adaptation of the Block and Kremen test. It consists of 12 questions assessed on a four-point scale, in which the answer 1 means “does not apply to me at all” and 4 “refers to me very strongly”. The overall score, which defines mental resilience, is obtained by summing up the partial scores from all the items. The tool also allows for assessment in two subscales: optimal regulation and openness to life experience. The tool has good psychometric properties, under which validity and reliability were tested. The reliability of the tool was assessed using Cronbach’s alpha coefficient and was 0.82 [14].

### 2.1. Satisfaction with Life Scale (SWLS)

The scale by Diener, Emmons, Larson and Griffin in the Polish adaptation of Juczyński allows for the study of life satisfaction understood as a subjective assessment of the quality of functioning. It contains five items. The respondent is asked to respond to each of the statements by specifying to what extent each of them applies to his/her life so far, from “strongly agree” (7 points) to “strongly disagree” (1 point). The grades are summed up and the result determines the level of satisfaction with life. Scores range from 5 to 35 points. They can be converted to standardized units on the sten scale. Scores in the range of 1–4 sten were considered low, and in the range of 7–10 sten were high. Scores in the range of 5–6 sten corresponded to average values. The reliability of the scale in Polish adaptation studies turned out to be good; Cronbach’s alpha coefficient was 0.81 [15,16].

### 2.2. Anxiety and Depression Scale (HADS)

The scale consists of two independent subscales containing 7 statements each, one of which assesses anxiety (HADS-A) and the other the severity of depression symptoms (HADS-D), taking into account the functioning of the respondent during the last week. Achieving 0–7 points in each of the subscales is considered normal [17].

The study was conducted online, during classes conducted in this form at both universities. Microsoft Forms were used in the study. The prepared form contained information for the participants about the purpose of the study, and the method of completing the tests used was discussed.

### 2.3. Statistical Analysis

Standard statistical procedures were used in the analyses. The Lilliefors-corrected Kolmogorov–Smirnov test was used to test the normality of distributions. The distributions of variables significantly deviated from the normal distribution, which is why it was decided to use nonparametric tests in further analyses. The Mann–Whitney U test was used to assess the significance of differences between the study groups. Spearman’s rank correlation coefficient was used to assess the relationships between the data. The significance level of *p* < 0.05 was assumed as statistically significant. Calculations were made in Statistica version 13.3.

## 3. Results

### 3.1. Ego-Resiliency

In the study group, the average score of 34.96 ± 5.19 points was obtained for medical students and 36.49 ± 5.22 points for students of the Academy of Physical Education; the difference was statistically significant (*p* = 0.003) (Table 2 and Table 3).

### 3.2. Satisfaction with Life

On the SWLS, an average score of 23.65 ± 5.9 points was obtained for medical students and 22.35 ± 5.67 points for students of the Academy of Physical Education; the difference was statistically significant (*p* = 0.005) (Table 2 and Table 3).

### 3.3. Anxiety and Depression Symptoms (HADS Scale)

In the anxiety assessment (HADS-A), medical students obtained about 8.43 ± 4.34 points and students of the Academy of Physical Education obtained 7.60 ± 4.27 points; the difference was statistically significant (*p* = 0.012). In the assessment of depression (HADS-D), medical students achieved 5.10 ± 3.77 points and students of APhE 4.77 ± 3.26 points; the difference was not statistically significant (Table 2 and Table 3).

### 3.4. Analysis of Relationships Between the Tested Parameters

The analysis of relationships between the examined parameters was presented for both study groups together. Significant, negative and moderate correlations were noted in terms of the results obtained on the ego-resiliency scale with the results obtained on the anxiety and depression scale. A positive and moderate correlation was described on the ego-resiliency scale with the results obtained on the SWLS scale (Table 4).

## 4. Discussion

The quality of life is conditioned by many factors. Some researchers emphasize that the relationship between the personality traits of an individual and the subjective sense of happiness may even be stronger than the relationship between psychological well-being and objective living conditions [18,19]. Ego-resiliency is among the human personality traits associated with the quality of life [20]. What is more, it can also be treated as a key factor in creating and implementing strategies for coping with stressful situations. Park et al. emphasize that the appropriate development of ego-resiliency can be a mechanism that compensates for the excessive emotional sensitivity of an individual [21]. The role of mental resilience is important at every stage of human development, fulfilling a special role in periods of increased stress. A study by Labrague et al. indicates the important role of ego-resiliency in adaptation mechanisms during the COVID-19 pandemic [22].

In this study, ego-resiliency among Polish medical students and students of the Academy of Physical Education was compared in a specific period of the COVID-19 pandemic. It had a multidimensional impact on mental health. Isolation, lack of social support and illnesses of one’s own or fear of death of a family member are just some of the examples. The situation related to the pandemic also led to job insecurity and fear of poverty. On this research, social support and socioeconomic situation were not explored, but the authors are aware of its importance. A statistically significant higher average score on the scale evaluating ego-resiliency was obtained in the group of students of the Academy of Physical Education in comparison to the students of medicine. Different results were obtained by Calo et al., who in their study obtained lower results of mental resilience in this study group. However, the study was conducted before the COVID-19 pandemic and included physiotherapy students only. Differences in ego-resilience may also result from the fact that it is also influenced by the conditions in which an individual was raised, social support and social relationships [23]. Research also shows that in the professional group of doctors, the level of occupational burnout reaches a high level, affecting people already during medical studies and appearing during professional work at various stages. The relationship between ER and burnout is bidirectional. Ego-resiliency is a personality trait that can help people deal with stress and avoid burnout, while burnout can be caused by low ego-resiliency [24,25]. Perhaps high workload at every stage of life with reduced involvement in pro-health activities, such as physical activity, in this professional group may hinder the development of ego-resiliency. In this study, in the assessment of life satisfaction, it was the medical students who achieved more points than the students of the APhE. Perhaps this was due to the fact that as people associated with healthcare units, they had greater opportunities to move freely during the pandemic; some classes were also held onsite, which gives better conditions for social functioning compared to people who held their classes mainly online. In addition, taking into account the fact that the group of people studying at the Academy of Physical Education is probably strongly predisposed to undertake physical activity, the periods of lockdown could be particularly severe for these people and could affect the assessment of life satisfaction. The lack of direct evidence for these explanations suggests the need of further research. The vast majority of studies conducted during the COVID-19 pandemic on the quality of life emphasize its worse assessment during this period, but none compared these two groups of students, as in our study [26,27]. In the assessment of depressive symptoms, the groups did not differ statistically, while the average score on the anxiety scale was statistically higher for medical students. This fact can be explained by the greater exposure of medical students to potential sources of infection with the SARS-CoV-2 virus compared to students of the Academy of Physical Education. Medical students often actively participated in helping COVID-19 patients in hospitals. In addition to the risk of infection, they also experienced death in a form and frequency they had not experienced before [28]. Moreover, in our study, by analyzing the relationship between the examined factors, which were estimated in both groups, positive correlations of ego-resiliency with life satisfaction and negative correlations with anxiety and depressive symptoms were obtained. A similar relationship was noted in the studies by both Seo et al. and Rospenda et al. [29,30]. Noreen came to interesting conclusions, emphasizing that the development of ego-resiliency should be especially promoted in the group of medical students in order to reduce the potential consequences associated with increased stress [31].

Summing up, it can be concluded that ego-resiliency may be a protective factor against symptoms of anxiety and depression and enhancing the quality of life. Some studies indicate that ego-resiliency as an internal human resource can be developed and modified throughout life [32,33]. Therefore, it would be beneficial to develop resilience from childhood, and especially during studies, when people usually already have a greater insight into the course of their mental processes. High ego-resiliency is a protective factor for an individual, thanks to which in difficult living conditions the individual can react in a less destructive way, without leading to anxiety or depression decompensation. Particularly useful methods could be psychotherapeutic techniques that arouse motivation to work on oneself, awakening awareness of one’s own emotions experienced in conflict situations, teaching to treat failures as challenges and stimulating students to overcome difficulties and change their behavior in the face of surprising or difficult situations.

## 5. Conclusions

Life during the pandemic was assessed by students of pro-health faculties as moderately satisfactory. The intensity of symptoms of anxiety and depression correlated negatively with life satisfaction. Mental resilience may be a protective factor against anxiety and depressive symptoms and may enhance life satisfaction.

## Figures and Tables

**Table 1 healthcare-13-01008-t001:** Sociodemographic characteristics of the group (n = 611).

	Total (n = 611)	Students of APhE (n = 249)	Medical Students (n = 362)	chi	*p*
	f (rf)	f (rf)	f (rf)		
Sex					
Male	220 (36.0%)	89 (35.7%)	131 (36.2%)	0.127	0.910
Female	391 (64.0%)	160 (64.3%)	231 (63.8%)		
Marital status					
Single	350 (57.2%)	141 (56.6%)	209 (57.7%)	0.074	0.786
In a relationship	261 (42.8%)	108 (43.4%)	153 (42.3%)		
Place of residence					
Urban	480 (78.6%)	209 (83.9%)	271 (74.9%)	7.211	0.007
Rural	131 (21.4%)	40 (16.1%)	91 (25.1%)		

**Table 2 healthcare-13-01008-t002:** Descriptive statistics of variables for the whole group.

N = 611	Mean	St. Dev.	Median	Min.	Max.
ER	35.584	5.689	36.000	32.000	40.000
SWLS	23.128	5.865	24.000	20.000	28.000
HADS-A	8.095	4.330	7.000	5.000	11.000
HADS-D	4.972	3.578	4.000	2.000	7.000

ER—Ego-resiliency, SWLS—The Satisfaction with Life Scale, HADS-A—The Hospital Anxiety and Depression Scale—Anxiety part, HADS-D—The Hospital Anxiety and Depression Scale—Depression part. St. Dev.—standard deviation.

**Table 3 healthcare-13-01008-t003:** Differences between medical students and students of the Academy of Physical Education (APhE).

	Medical Students (n = 362)			Students of APhE (n = 249)		Mann–Whitney U
	Mean	St. Dev.	Median	Mean	St. Dev.	*p*
ER *	34.961	5.913	35.000	36.490	5.226	0.003
SWLS *	23.657	5.940	25.000	22.357	5.679	0.005
HADS-A *	8.434	4.345	8.000	7.602	4.270	0.012
HADS-D	5.105	3.777	4.000	4.779	3.266	0.13

ER—Ego-resiliency, SWLS—The Satisfaction with Life Scale, HADS-A—The Hospital Anxiety and Depression Scale—Anxiety part, HADS-D—The Hospital Anxiety and Depression Scale—Depression part. St. Dev.—standard deviation. * statistically significant *p* < 0.05.

**Table 4 healthcare-13-01008-t004:** Relationships between ego-resiliency, life satisfaction and anxiety and depressive symptoms.

n = 611	ER	SWLS	HADS-A	HADS-D
ER	1.000	0.467 **	−0.390 **	−0.436 *
SWLS		1.000	−0.466 **	−0.572 *
HADS-A			1.000	0.633 ***
HADS-D				1.000

ER—Ego-resiliency, SWLS—The Satisfaction with Life Scale, HADS-A—The Hospital Anxiety and Depression Scale—Anxiety part, HADS-D—The Hospital Anxiety and Depression Scale—Depression part. * statistically significant *p* < 0.05, ** *p* < 0.01, *** *p* < 0.001.

## Data Availability

The original contributions presented in this study are included in the article. Further inquiries can be directed to the corresponding author (Patryk Główczyński).

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
