# Peer review of "Ego-Resiliency, Life Satisfaction and Symptoms of Anxiety and Depression Among Students of Pro-Health Faculties During COVID-19 Pandemic"

_healthcare, 2025, doi:10.3390/healthcare13091008_

Round 1
Reviewer 1 Report
Comments and Suggestions for Authors
The aim of the study was to assess ego-resiliency as a potential protective factor 90
against anxiety and depressive symptoms and having a positive relationship with life sat- 91
isfaction in a group of health students during the SARS-CoV-2 virus pandemic.
The paper is interesting, however, some clarification/modifications are needed for the paper
What is the significance of using Mann Whitney and Spearman's rank correlation? you did not mention the distribution of data?
the correlation modes needs to be adjusted to control for gender and other demographics.
there is no discussion about confounding variables that could impact your results.
Author Response
Dear Reviewer,
Here are our answers for your questions:
The aim of the study was to assess ego-resiliency as a potential protective factor 90 against anxiety and depressive symptoms and having a positive relationship with life satisfaction in a group of health students during the SARS-CoV-2 virus pandemic.
The paper is interesting, however, some clarification/modifications are needed for the paper
What is the significance of using Mann Whitney and Spearman's rank correlation? you did not mention the distribution of data?
The Lilliefors-corrected Kolmogorov-Smirnov test was used to test the normality of distributions. The distributions of variables significantly deviated from the normal distribution, which is why it was decided to use nonparametric tests in further analyses.
The correlation modes needs to be adjusted to control for gender and other demographics.
Additional correlation matrices were prepared, taking into account the division of the study group according to socio-demographic variables. Due to limited space, correlation tables are placed in supplementary materials.
there is no discussion about confounding variables that could impact your results.
It was done in discussion.
Thank you for your work.
Sincerely,
Authors
Reviewer 2 Report
Comments and Suggestions for Authors
- Main Question Addressed:
The main question addressed by the research is, essentially: "How does ego-resiliency relate to life satisfaction, anxiety, and depression in students of pro-health faculties during the COVID-19 pandemic?" More specifically, the authors are interested in whether ego-resiliency acts as a protective factor against anxiety and depression and a positive correlate of life satisfaction in this population during a period of significant stress.
The aim is explicitly mentioned. Lines 16-19 in the abstract state: The aim of the study was to assess ego-resiliency as a factor enhancing life satisfaction and a protective factor against symptoms of anxiety and depression in a group of health students during COVID-19 pandemic.. A second aim mention appears at line 89-92.
- Originality and Relevance:
- Originality: While studies on resilience, life satisfaction, and mental health during the pandemic are numerous, this study has several original aspects:
- Specific Population: It focused on students of "pro-health faculties," specifically comparing medical students with students from the Academy of Physical Education (APhE). This is a novel comparison group, as most studies focus solely on medical students or general student populations.
- Timing: The data collection occurred during a specific period of the pandemic (October 2020 to June 2021). This provided a snapshot of a particular phase of the pandemic, which might differ from studies conducted earlier or later.
- The comparison with APhE students made this work valuable.
- Relevance and Gap Addressed: The paper addressed a relevant gap by investigating the mechanisms through which individuals cope with stress during a global crisis. It went beyond simply documenting the prevalence of anxiety and depression (which many studies have done) and explored the role of a specific personality trait (ego-resiliency) in mediating these outcomes. This contributed to our understanding of how to build resilience-promoting interventions, specifically in populations potentially more vulnerable due to their professional training (healthcare and, interestingly, physical education). The paper connected well with the literature by the correct selection of Block´s theory of Ego resiliency, and all of the instruments selected have been probed and tested.
- Methodological Improvements:
The following suggestions could strengthen the methodological rigour of the study:
- 3.1 Sample Size and Sampling:
- The provided document gave limited information on sample collection. Was a confidence Interval and confidence level calculation applied before collecting the sample? A total of 611 completed the surveys, but there is no answer on whether this is an adequate number to avoid Type II error.
- Recruitment (Lines 94-97 & 132-135): The description of recruitment is very brief: "All respondents agreed to participate in the project" and "The study was conducted online, during classes conducted in this form at both universities.". This raises several questions:
- How were students invited to participate? Was it a general announcement, or were specific classes targeted? This could introduce selection bias.
- What was the response rate? Knowing what proportion of eligible students agreed to participate is crucial for assessing the representativeness of the sample. A low response rate could significantly bias the results.
- Were students incentivised in any way? Incentives can influence who chooses to participate.
- Was the online data collected in class synchronic or anachronic? There is no further mention on weather students were at home or school doing this.
- Representativeness: The manuscript did not adequately discuss the representativeness of the sample to the broader population of medical and APhE students in Poland, or beyond. Are these two universities typical? Were students from all years of study included, or only certain years? This limits generalisability.
- 3.2. Study Design:
- Cross-Sectional Design: The study used a cross-sectional design, meaning data were collected at a single point in time. This makes it impossible to determine causality. We cannot conclude that ego-resiliency causes lower anxiety or higher life satisfaction; it could be the other way around, or a third variable could influence all three. A longitudinal design, tracking students over time, would be far stronger for examining these relationships. This weakness should be stated earlier.
- 3.3. Materials:
- The questionnaires are good. No further considerations on this.
- 3.4. Controls and Confounding Variables:
- Socioeconomic Status (SES): SES is a significant potential confounder, as it is linked to both mental health outcomes and access to resources that could buffer stress. The study did not appear to control for SES. This omission should be addressed, ideally through including SES as a covariate in analyses, and is in odds with lines 221-223, that indicates that financial resources are important.
- Prior Mental Health History: Pre-existing mental health conditions (before the pandemic) would strongly influence anxiety and depression levels during the pandemic. The study did not assess this, which is a significant limitation.
- Social Support: While social support is discussed in the introduction as a component of resilience, it was not directly measured in the study, and it is a crucial factor to consider, particularly during periods of isolation. The absence of a social support measure (e.g., the Multidimensional Scale of Perceived Social Support) is a weakness. Any take on that?
- COVID-19 Specific Stressors: The study mentioned the pandemic as a general stressor, but it did not quantify specific COVID-19 related stressors (e.g., personal illness, illness of family members, financial hardship due to the pandemic, academic disruption). These specific stressors could differentially impact students and should be considered.
- Coping Strategies: Line 84 mentioned this aspect as important, but there is no further research.
- Consistency of Conclusions:
The conclusions are generally consistent with the evidence presented, given the limitations of the cross-sectional design.
- Evidence for Correlations: The study found significant correlations between ego-resiliency, life satisfaction, anxiety, and depression, as hypothesised. The negative correlations between ego-resiliency and anxiety/depression, and the positive correlation with life satisfaction, support the idea of ego-resiliency as a protective factor.
- Causality: The authors were relatively cautious in their language, using phrases, such as "may be a protective factor" (Lines 36-37), which was appropriate. However, phrases, such as "enhancing life satisfaction" (Line 36) could be interpreted as implying causality, which was not supported by the cross-sectional data. The manuscript should explicitly state the limitations regarding causal inferences.
- Addressing the Main Question: The main question (the relationship between ego-resiliency and mental health outcomes during the pandemic) was addressed through the correlational analyses. The comparison between medical and APhE students adds an additional layer of analysis, although the reasons for the observed differences remain speculative.
- Tables and Figures:
- Table 1: Well-presented and informative. It provides a good overview of the demographic characteristics of the two groups. It might be helpful to add a column showing p-values for comparisons between the groups on these demographic variables (e.g., chi-square tests for categorical variables, t-tests or Mann-Whitney U tests for continuous variables).
- Table 2: Clearly presents the descriptive statistics for the whole group.
- Table 3: This is a key table, showing the group comparisons. The use of the Mann-Whitney U test is appropriate given the likely non-normal distribution of the data. The presentation is clear.
- Table 4: The use of Spearman's rank correlation is appropriate. The presentation is clear, and the use of asterisks to denote significance levels is standard. The strength of the relationship must be defined.
- Caveats, Weaknesses, and Mistakes:
- Introduction:
- Definition of "Pro-Health Faculties" (Line 3): The term "pro-health faculties" is not a standard term and requires immediate clarification. It is crucial to explain what types of faculties are included in this category. The later distinction between medical students and APhE students helps, but the initial term remains vague.
- Lines 45 to 46. Consider adding that Block refers as ego-control the tendency to contain versus express impulse, and ego-resiliency as the ability to modulate, in either direction, his or her characteristic level of ego-control, depending of situational demands. Adding definitions and descriptions in academic publications have a strong benefit for unexperienced readers.
- Methods:
- See the extensive comments in section 3 above regarding limitations of sampling, design, and controls.
- Line 108. Clarify better popular abroad, what that means?
- The anxiety assessment (HADS-A) needs the range value that are normal (Lines 151-156)
- Discussion:
- Overgeneralisation: The discussion sometimes generalises findings to "students" broadly, without acknowledging that the sample was limited to students from specific types of faculties in Poland.
- Speculation Without Sufficient Support: Some of the explanations for the group differences (e.g., Lines 213-219 regarding why APhE students had lower life satisfaction) are plausible but speculative. The authors should acknowledge the lack of direct evidence for these explanations and suggest further research.
- Line 221-223, needs some review with line 84. Both sections do not match properly.
- Repetitive Discussion Points: Some points are repeated across the discussion, making it feel somewhat redundant. A more concise and focused discussion would be stronger.
- Line 228-231 needs citations to reinforce the presented ideas.
- Contradictory interpretations. There are at least 2 ideas, regarding why ego resiliency scored less on Medical Student, that are on the table. Both should be considered more thoroughly.
Author Response
Dear Reviewer,
Here are the answers for your questions and suggestions.
- Methodological Improvements:
The following suggestions could strengthen the methodological rigour of the study:
- 3.1 Sample Size and Sampling:
- The provided document gave limited information on sample collection. Was a confidence Interval and confidence level calculation applied before collecting the sample? A total of 611 completed the surveys, but there is no answer on whether this is an adequate number to avoid Type II error.
930 students were invited to participate in the study. The study was conducted using university e-learning platforms. Considering that the number of students who could take part in the study was estimated at 930, then assuming the size of the fraction = 0.5, the maximum error of 2% and α = 0.95, the sample size should be 670 people. Assuming a maximum error of 5%, this number is 272 people.
- Recruitment (Lines 94-97 & 132-135): The description of recruitment is very brief: "All respondents agreed to participate in th
- e project" and "The study was conducted online, during classes conducted in this form at both universities.". This raises several questions:
- How were students invited to participate? Was it a general announcement, or were specific classes targeted? This could introduce selection bias. Students were encouraged by the lecturers, i.e. the authors of the study. It was based on a general announcement.
- What was the response rate? Knowing what proportion of eligible students agreed to participate is crucial for assessing the representativeness of the sample. A low response rate could significantly bias the results. Those who were willing filled in the questionnaire on-site, using phones and tablets. Before the participation, students signed the special agreement form. 93% of invited students filled in the questionnaires.
- Were students incentivised in any way? Incentives can influence who chooses to participate. There was not special incentives although students got information about the importance of the research and the special role of students in that analysis.
- Was the online data collected in class synchronic or anachronic? There is no further mention on weather students were at home or school doing this. Data collection took place synchronously in student groups after the lectures.
- Representativeness: The manuscript did not adequately discuss the representativeness of the sample to the broader population of medical and APhE students in Poland, or beyond. Are these two universities typical? Were students from all years of study included, or only certain years? This limits generalisability. Medical University of Silesia and Academy of Physical Education (APhE) in Katowice were typical universities, whatsmore, for that time (pandemic) on Silesia Voivodeship there were the only universities which educate people on that field of studies.
- 3.2. Study Design:
- Cross-Sectional Design: The study used a cross-sectional design, meaning data were collected at a single point in time. This makes it impossible to determine causality. We cannot conclude that ego-resiliency causes lower anxiety or higher life satisfaction; it could be the other way around, or a third variable could influence all three. A longitudinal design, tracking students over time, would be far stronger for examining these relationships. This weakness should be stated earlier. We added this to conclusions. Thank you for your valuable note.
- 3.3. Materials:
- The questionnaires are good. No further considerations on this.
- 3.4. Controls and Confounding Variables:
- Socioeconomic Status (SES): SES is a significant potential confounder, as it is linked to both mental health outcomes and access to resources that could buffer stress. The study did not appear to control for SES. This omission should be addressed, ideally through including SES as a covariate in analyses, and is in odds with lines 221-223, that indicates that financial resources are important. We added this to study limitations in a point of discussion. Thank you for your valuable note.
- Prior Mental Health History: Pre-existing mental health conditions (before the pandemic) would strongly influence anxiety and depression levels during the pandemic. The study did not assess this, which is a significant limitation. We added this to study limitations and a point in a discussion. Thank you for your valuable note.
- Social Support: While social support is discussed in the introduction as a component of resilience, it was not directly measured in the study, and it is a crucial factor to consider, particularly during periods of isolation. The absence of a social support measure (e.g., the Multidimensional Scale of Perceived Social Support) is a weakness. Any take on that? We added this to study limitations and a point in a discussion. Thank you for your valuable note.
- COVID-19 Specific Stressors: The study mentioned the pandemic as a general stressor, but it did not quantify specific COVID-19 related stressors (e.g., personal illness, illness of family members, financial hardship due to the pandemic, academic disruption). These specific stressors could differentially impact students and should be considered. We added this to study limitations. Thank you for your valuable note.
- Coping Strategies: Line 84 mentioned this aspect as important, but there is no further research. We added this to study limitations and a point in a discussion. Thank you for your valuable note.
- Consistency of Conclusions:
The conclusions are generally consistent with the evidence presented, given the limitations of the cross-sectional design.
- Causality: The authors were relatively cautious in their language, using phrases, such as "may be a protective factor" (Lines 36-37), which was appropriate. However, phrases, such as "enhancing life satisfaction" (Line 36) could be interpreted as implying causality, which was not supported by the cross-sectional data. The manuscript should explicitly state the limitations regarding causal inferences. We change the sentences for „may enhance life satisfaction’’
- Tables and Figures:
- Table 1: Well-presented and informative. It provides a good overview of the demographic characteristics of the two groups. It might be helpful to add a column showing p-values for comparisons between the groups on these demographic variables (e.g., chi-square tests for categorical variables, t-tests or Mann-Whitney U tests for continuous variables).
We examined the differences in socio-demographic variables between the study groups using the chi-square test. The data are presented in Table 1.
- Table 4: The use of Spearman's rank correlation is appropriate. The presentation is clear, and the use of asterisks to denote significance levels is standard. The strength of the relationship must be defined.
We supplemented the description of correlations with the strength of relationships – they were all moderate. For this purpose, we adopted the classification proposed by Guilford (Van Aswegen, A.S. & Engelbrecht, A.S. 2009. The relationship between transformational leadership, integrity and an ethical climate in organisations. SA Journal of Human Resource Management/SA Tydskrif vir Menslikehulpbronbestuur, 7(1); Art. #175, doi:10.4102/sajhrm.v7i1.175.)
- Caveats, Weaknesses, and Mistakes:
- Introduction:
- Definition of "Pro-Health Faculties" (Line 3): The term "pro-health faculties" is not a standard term and requires immediate clarification. It is crucial to explain what types of faculties are included in this category. The later distinction between medical students and APhE students helps, but the initial term remains vague. We added clarification in the background, thank you.
- Lines 45 to 46. Consider adding that Block refers as ego-control the tendency to contain versus express impulse, and ego-resiliency as the ability to modulate, in either direction, his or her characteristic level of ego-control, depending of situational demands. Adding definitions and descriptions in academic publications have a strong benefit for unexperienced readers. We added definition, thank you for that idea.
- Methods:
- See the extensive comments in section 3 above regarding limitations of sampling, design, and controls.
- Line 108. Clarify better popular abroad, what that means? We change the sentence.
- The anxiety assessment (HADS-A) needs the range value that are normal (Lines 151-156) We added the sentence about it.
- Discussion:
- Overgeneralisation: The discussion sometimes generalises findings to "students" broadly, without acknowledging that the sample was limited to students from specific types of faculties in Poland. We tried to change the discussion to eliminate the generalisation, added new information about limitations.
- Speculation Without Sufficient Support: Some of the explanations for the group differences (e.g., Lines 213-219 regarding why APhE students had lower life satisfaction) are plausible but speculative. The authors should acknowledge the lack of direct evidence for these explanations and suggest further research. - We added the sentences in the discussion.
- Line 221-223, needs some review with line 84. Both sections do not match properly. - Done
- Repetitive Discussion Points: Some points are repeated across the discussion, making it feel somewhat redundant. A more concise and focused discussion would be stronger. - Discussion was changed and shortened.
- Line 228-231 needs citations to reinforce the presented ideas. - Done
- Contradictory interpretations. There are at least 2 ideas, regarding why ego resiliency scored less on Medical Student, that are on the table. Both should be considered more thoroughly. - Differences in ego-resilience may also result from the fact that it is also influenced by the conditions in which an individual was raised, social support and social relationships. Research also shows that in the professional group of doctors, the level of occupational burnout reaches a high level, affecting people already during medical studies and appearing during professional work at various stages. The relationship between ER and burnout is bidirectional. Ego-resiliency is a personality trait that can help people deal with stress and avoid burnout, while burnout can be caused by low-ER. Perhaps high workload at every stage of life with reduced involvement in pro-health activities, such as physical activity, in this professional group may hinder the development of ego-resiliency.
We would like to thank for your work.
Sincerely,
Authors
Reviewer 3 Report
Comments and Suggestions for Authors
The COVID-19 pandemic has had a significant negative impact on mental health worldwide. Ego-resiliency (ER) is a set of personality traits that facilitate positive adaptation to stress, potentially serving as a protective factor during the pandemic. This study investigated the effects of ego-resiliency on psychological distress associated with COVID-19 among medical students who volunteered to collect specimens and treat patients. The finding suggest that ego-resiliency helped these students manage anxiety and depression during the pandemic. While this manuscript provides valuable insights, several key issues remain to be addressed.
Major comments:
- The authors compared the ER, SWLS and anxiety/depression symptom severity between medical students and students at APhE. As shown in table 1, these two groups differ in several aspects beyond their ER score, such as place of residence and chronic diseases. Notably, the incidence of chronic diseases among medical students was approximately twice as high as in students in students at APhE (31.1% vs. 16%). How do the authors account for the potential influence of these factors on SWLS and anxiety/depression symptom scores?
- Given the various differences between medical students and students at APhE, it would be insightful to analyze the ER scale, SWLS points and anxiety/depression severity within medical student group to further examine the relationship between ER and these variables.
- Line 170: The authors stated, “Positive correlations were described in the ego-resiliency scale with the 170 results obtained in the SWLS scale.” However, medical students, who had lower ER scores (34.96 ± 5.19 vs. 36.49 ± 5.22), exhibited higher SWLS (23.65 ± 5.9) scores than students of APhE (22.35 ± 5.67). How do the authors explain this contradiction?
- The manuscript references previous studies (line 195 - 198) that reported lower ER scores in student at APhE. The authors attributed these differences to the fact that prior studies were conducted before the pandemic. Does this suggest that the COVID-19 pandemic influenced students’ ER levels? If so, how?
- How do the authors establish a causal relationship between high ER and lower anxiety/depression symptom severity? Could both be consequences of the stress induced by the pandemic rather than one directly influencing the other?
Minor comments:
Line 34: “the results obtained in the SWLS scale for both groupS”. “S” should be lowercase.
Line 91: the word “satisfaction” should be hyphenated correctly.
Line 115 and 126: The manuscript inconsistently refers to Cronback’s É‘ coefficient as Greek letter “É‘” (Line 115) and English word “alpha” (Line 126). Please ensure consistency in the formatting.
Author Response
Dear Reviewer,
Thank your for your valuable impact on our work. Here are the answers for your questions.
Major comments:
The authors compared the ER, SWLS and anxiety/depression symptom severity between medical students and students at APhE. As shown in table 1, these two groups differ in several aspects beyond their ER score, such as place of residence and chronic diseases. Notably, the incidence of chronic diseases among medical students was approximately twice as high as in students in students at APhE (31.1% vs. 16%). How do the authors account for the potential influence of these factors on SWLS and anxiety/depression symptom scores? We added that information on the discussion for a wider perspective, thank you for your idea.
Given the various differences between medical students and students at APhE, it would be insightful to analyze the ER scale, SWLS points and anxiety/depression severity within medical student group to further examine the relationship between ER and these variables. - Correlation matrices taking into account the relationships between the indicated variables in various groups distinguished on the basis of socio-demographic variables, including the group of medical students, were included in the supplementary materials.
Line 170: The authors stated, “Positive correlations were described in the ego-resiliency scale with the 170 results obtained in the SWLS scale.” However, medical students, who had lower ER scores (34.96 ± 5.19 vs. 36.49 ± 5.22), exhibited higher SWLS (23.65 ± 5.9) scores than students of APhE (22.35 ± 5.67). How do the authors explain this contradiction? - The observed differences may indicate a different psychological situation of students representing these two fields of study. Higher SWLS intensity in medical students may be related to many factors, including those related to their future profession - such as employment stability, good salary, specific profession and skills, the possibility of fulfilling prosocial and humanitarian needs, high social position and respect. At the same time, positive relationships between ER and SWLS were revealed in the general group, which was expected and generally indicates that these two factors may be associated with mutual growth. However, it should be remembered that these are only correlations, and life satisfaction (SWLS) may result from many different psychosocial conditions, not only from ER.
The manuscript references previous studies (line 195 - 198) that reported lower ER scores in student at APhE. The authors attributed these differences to the fact that prior studies were conducted before the pandemic. Does this suggest that the COVID-19 pandemic influenced students’ ER levels? If so, how? - We changed that and made new assumptions: Differences in ego-resilience may also result from the fact that it is also influenced by the conditions in which an individual was raised, social support and social relationships.
How do the authors establish a causal relationship between high ER and lower anxiety/depression symptom severity? Could both be consequences of the stress induced by the pandemic rather than one directly influencing the other? - High ego-resiliency is a protective factor of an individual, thanks to which in difficult living conditions the individual can react in a less destructive way, without leading to anxiety or depression decompensation. People with high ego-resiliency are better able to cope with stress, trauma, and setbacks. What is important, perceiving high stress level doesn’t have to make high resilience.
Minor comments:
Line 34: “the results obtained in the SWLS scale for both groupS”. “S” should be lowercase. - done
Line 91: the word “satisfaction” should be hyphenated correctly. - done
Line 115 and 126: The manuscript inconsistently refers to Cronback’s É‘ coefficient as Greek letter “É‘” (Line 115) and English word “alpha” (Line 126). Please ensure consistency in the formatting. - done
Sincerely,
Authors
Round 2
Reviewer 2 Report
Comments and Suggestions for Authors
Glad with changes